# A half-century of changes in migratory landbird numbers along coastal Massachusetts

**Matthew D. Kamm** [1]*, **Trevor L. Lloyd-Evans**[2], **Maina Handmaker**[2], **J. Michael Reed**[1]

**1** Department of Biology, Tufts University, Medford, Massachusetts, United States of America, **2** Manomet, Plymouth, Massachusetts, United States of America

* matthew.kamm@tufts.edu

**Data Availability Statement:** All data used in these analyses may be accessed in the Supplemental Material, organized by season in the files S2 and S3 Tables.

## Abstract

We analyzed data from across five decades of passerine bird banding at Manomet in Plymouth, Massachusetts, USA. This included 172,609 captures during spring migration and 253,265 during fall migration, from 1969 to 2015. Migration counts are prone to large inter-annual variation and trends are often difficult to interpret, but have the advantage of sampling many breeding populations in a single locale. We employed a Bayesian state-space modeling approach to estimate patterns in abundance over time while accounting for observation error, and a hierarchical clustering method to identify species groups with similar trends over time. Although continent-wide there has been an overall decrease in landbird populations over the past 40 years, we found a variety of patterns in abundance over time. Consistent with other studies, we found an overall decline in numbers of birds in the aggregate, with most species showing significant net declines in migratory cohort size in spring, fall, or both (49/73 species evaluated). Other species, however, exhibited different patterns, including abundance increases (10 species). Even among increasing and declining species, the specific trends varied in shape over time, forming seven distinct clusters in fall and ten in spring. The remaining species followed largely independent and irregular pathways. Overall, life-history traits (dependence on open habitat, nesting on or near the ground, migratory strategy, human commensal, spruce budworm specialists) did a poor job of predicting species groupings of abundance patterns in both spring and fall, but median date of passage was a good predictor of abundance trends during spring (but not fall) migration. This suggests that some species with very similar patterns of abundance were unlikely to be responding to the same environmental forces. Changes in abundance at this banding station were generally consistent with BBS trend data for the same geographic region.

## Introduction

Long-term data sets in ecology lead to discoveries often missed in shorter-term studies [1,2], and they are critical for establishing baselines and tracking changes in the natural world [3]. Because birds are widely surveyed by professional and amateur observers alike, and their

**Funding:** This work was funded entirely by the generosity of Manomet's friends, donors, and trustees. This funding paid the salaries of MK, MH, and TLE during their work on this project. The donors had no role in study design, data collection and analysis, decision to publish, or preparation of the manuscript. All data collection was done by Manomet employees or volunteers; design of this study, data analysis, decision to publish, and preparation of the manuscript were done by the authors with no input from any donor or trustee.

**Competing interests:** The authors have declared that no competing interests exist.

natural histories are often well-understood, wild bird populations can be useful sentinels of environmental change and ecosystem condition [4,5] For example, during the 1980s and 1990s, wide-spread surveys were used to identify large-scale declines of birds across the continental U.S. and Canada [6–8]. Contemporary interests include documenting species' range shifts due to climate change [9–11], and modeling the spread of exotic, invasive species like Eurasian Collared-Doves *Streptopelia decaocto* [12]. Prominent long-term bird monitoring data in North America are available for breeding birds in the USGS Breeding Bird Survey (BBS) [13] and the Christmas Bird Count (CBC) [14]. These surveys amass a wealth of valuable data on bird abundance, but there are biases and gaps in survey coverage that necessitate the integration of other data sources. For example, BBS data are biased in space because they are roadside surveys [15,16], and by being of short count duration [17,18]. Survey gaps can be temporal (e.g., during migration) or spatial (e.g., off-shore rocky islands), which in turn makes certain taxa much less likely to be detected. These gaps are filled by other monitoring programs. For example, eBird is allowing large-scale identification of migratory stopover and wintering areas [19], as are targeted taxon surveys such as the International Shorebird Survey (ISS) [20] and various hawk migration watches [21].

Migratory bird banding operations represent an underutilized source of data about the stages of avian life that connect breeding and wintering: migration [22]. These sites often have long-term datasets collected by highly-trained observers, coupled with detailed data on capture effort and local conditions. Although migratory bird data from a single banding station should be interpreted with care because of yearly stochasticity introduced by fluctuations in weather conditions [23,24], banding stations identified the previously-unknown breeding grounds of wintering sparrows in California [25], and demonstrated differences in stopover energetics between hatch-year and adult birds in southern Canada [26]. In addition to answering basic questions of natural history, banding station data have recently been used to describe and assess the way migrating birds are responding to climate change, both in Europe [27] and the Americas [28]. Because banding stations are typically situated in areas such as mountain gaps, desert oases, and coastal points where birds from many breeding populations naturally aggregate during migration, they can evaluate changes in population size across a much larger region from a typical single point in space [22].

Our goal was to analyze fall and spring migration banding data for >50 species of landbirds across almost half a century from Manomet, a banding station in eastern North America, along the Atlantic Flyway. There has been a series of important studies looking at population changes of breeding and migrating birds in this region of North America, mostly focusing on population declines and changing migration phenology associated with global climate change, including extensive work at Manomet [6,7, 29–32]. Often ignored, however, is the presence of stable and increasing species (e.g., Blue-headed Vireo *Vireo solitarius*, Carolina Wren *Thryothorus ludovicianus*) [13,33], perhaps because the stories are less dramatic. Previous analyses of Manomet migration data have focused on attempting to identify common trends among Neotropical migrant species [30,31]. We aim to take these analyses a step further, first by expanding the database with over a decade of new data, and then by examining the ways in which migratory bird abundances cluster over time. We then attempt to quantitatively evaluate what life-history factors best predict these groupings. By using a Bayesian state-space approach to analyzing migration count data, we use more accurate estimates of real trends in migratory cohort size to reduce the uncertainty in identifying species clusters. Generalizations are often made about the particular population vulnerability of, for example, birds of grassland and agricultural habitats [34] or birds that winter in the Neotropics [6]. If these life-history traits and their associated risks are driving the declines of avian guilds, then the shapes of population trends within that guild should be generally similar. This approach allows us to examine

whether life-history traits predict observed trends in migratory cohort size, and to better iden-
tify species that are doing especially well or especially poorly along with mechanisms for the
different patterns of change.

## Methods

Manomet's banding lab has operated mist nets at their coastal site in Plymouth, Massachusetts,
USA (41° 50' N, 70° 30' W) every spring and fall since 1966. The site is dominated by second-
growth hardwoods, but also borders swampy areas, old cleared fields, and a seaside coastal
bluff. Migrating birds were captured using a system of 45–50 (depending on year) nylon mist
nets (12 m long, 2.6 m high, 36 mm extended mesh) set at fixed spots along a series of trails
covering part of the Manomet property. Opening and closing times for all nets were recorded
and used to create a standard effort measure of net-hours. Nets were typically open from a
half-hour before sunrise to a half-hour after sunset, 5–7 days per week depending on weather,
in spring (15 April– 15 June) and fall (15 August– 15 November). Nets were occasionally
closed due to weather conditions that might endanger birds; all such closures were recorded
and factored into effort calculations.

All banding activity at Manomet was performed by trained personnel under the supervision
of a master bander with an active Federal Bird Banding and Marking Permit from the USGS
Bird Banding Lab, and MassWildlife bird banding and salvage permits from the Massachusetts
Division of Fisheries and Wildlife. Lloyd-Evans is also a bird banding trainer certified by the
North American Banding Council (1998).

Four-letter abbreviation codes and scientific names for all species analyzed appear in
Table 1. Relevant wing formula data for the separation of Alder Flycatcher and Willow Fly-
catcher were not collected for more than half the study period, thus we have adopted the parsi-
monious strategy of not separating these two species trends and both are presented as "Traill's
Flycatcher.".. Subspecies of Palm Warbler were recorded as Yellow Palm Warblers (*Setophaga
palmarum hypochrysea*) and Western Palm Warbler (*S. p. palmarum*). Hybrid Blue-winged x
Golden-winged Warblers were recorded as Blue-winged Warblers. For a handful of species
frequently caught in ground traps (White-throated Sparrow, Red-winged Blackbird, and
Brown-headed Cowbird), we included hours of ground trap deployment in calculations of
total effort-hours.

### Data processing

Records used in this analysis were from 1969–2015, and excluded repeat captures within the
same season as well as local breeders (distinguished by physiological signs of breeding readi-
ness, or local fledgling birds caught during spring migration). Although Manomet personnel
banded birds from over 150 species during the target years, many of these occurred only a
handful of times. Within fall and spring data, we removed from analysis all species not caught
in at least 15 different years, and then examined the total birds caught for each of the remain-
ing species. For the 108 remaining species, we removed those that had insufficient data to
determine a significant trend, which we defined as a capture rate <5 individuals/year. This left
us with 62 species with sufficient data to be analyzed for fall migration trends, and 52 species
for spring migration.

Because sampling effort varies through time, and different species have their migratory
peaks at different parts of each season, we calculated a peak migration window for each species
according to the methods used in Lloyd-Evans and Atwood [31]. Briefly, we identified the
dates that accounted for 98% of all captures—thereby excluding the first and last 1% of cap-
tures—for each species across all years, and excluded sampling effort outside this period when

**Table 1. Complete summary of all species analyzed.**

| Species | Code | Cluster+ | | Number Caught | | Overall Trends | | BBS Trends+ | | |
|---|---|---|---|---|---|---|---|---|---|---|
| | | Fall | Spr | Fall | Spr | Fall | Spr | NEMA | ANF | BSS |
| Ruby-throated Hummingbird (*Archilochus colubris*) | RTHU | – | – | – | 837 | NA | Increase | Increase | Increase | – |
| Downy Woodpecker (*Dryobates pubescens*) | DOWO | – | – | 844 | – | Stable | NA | Increase | Increase | Stable |
| Northern (Yellow-shafted) Flicker (*Colaptes a. auratus*) | YSFL | – | – | 607 | – | Decline | Decline | Decline | Decline | Decline |
| Eastern Wood-Pewee (*Contopus virens*) | EAWP | – | 6 | 129 | 264 | NA | Decline | Stable | Decline | – |
| Yellow-bellied Flycatcher (*Empidonax flaviventris*) | YBFL | 1 | – | 261 | 526 | Decline | Decline | – | Stable | Increase |
| Alder & Willow (Traill's) Flycatcher (*E. alnorum & traillii*) | TRFL | – | 1 | 445 | 1442 | Decline | Decline | Increase | Stable | Stable |
| Least Flycatcher (*E. minimus*) | LEFL | – | 2 | 395 | 419 | Decline | Decline | Decline | Decline | Decline |
| Eastern Phoebe (*Sayornis phoebe*) | EAPH | 4 | – | 735 | 217 | Increase | Stable | Stable | Decline | Stable |
| Great Crested Flycatcher (*Myiarchus crinitus*) | GCFL | – | – | – | 422 | NA | Increase | Increase | Decline | – |
| Blue-headed Vireo (*Vireo solitarius*) | BHVI | 4 | – | 484 | – | Increase | NA | Stable | Increase | Increase |
| Philadelphia Vireo (*V. philadelphicus*) | PHVI | – | – | 244 | – | Stable | NA | – | Increase | Increase |
| Red-eyed Vireo (*V. olivaceus*) | REVI | 5 | – | 3627 | 464 | Decline | Decline | Decline | Increase | Increase |
| Blue Jay (*Cyanocitta cristata*) | BLJA | 6 | 10 | 2352 | 3211 | Decline | Decline | Decline | Increase | Increase |
| Black-capped Chickadee (*Poecile atricapillus*) | BCCH | 2 | | 29064 | 1255 | Decline | Decline | Stable | Increase | Increase |
| Tufted Titmouse (*Baeolophus bicolor*) | TUTI | – | 5 | 5814 | 399 | Increase | Increase | Increase | Increase | – |
| Red-breasted Nuthatch (*Sitta canadensis*) | RBNU | 3 | – | 230 | – | Decline | NA | Stable | Increase | Stable |
| White-breasted Nuthatch (*S. carolinensis*) | WBNU | – | – | 292 | – | Increase | NA | Increase | Increase | – |
| Brown Creeper (*Certhia americana*) | BRCR | 1 | 6 | 1677 | 192 | Decline | Decline | Stable | Increase | Increase |
| Carolina Wren (*Thryothorus ludovicianus*) | CARW | – | – | 599 | – | Increase | NA | Increase | – | – |
| Winter Wren (*Troglodytes hiemalis*) | WIWR | 1 | – | 206 | – | Decline | NA | – | Stable | Stable |
| Golden-crowned Kinglet (*Regulus satrapa*) | GCKI | 1 | – | 3090 | – | Decline | NA | – | Increase | Stable |
| Ruby-crowned Kinglet (*R. calendula*) | RCKI | 5 | 9 | 1921 | 1245 | Decline | Decline | – | Decline | Stable |
| Blue-gray Gnatcatcher (*Polioptila caerulea*) | BGGN | 7 | – | 341 | – | Stable | NA | Increase | – | – |
| Veery (*Catharus fuscescens*) | VEER | – | – | 561 | 612 | Decline | Stable | Decline | Decline | Increase |
| Swainson's Thrush (*C. ustulatus*) | SWTH | – | – | 1255 | 1517 | Stable | Decline | – | Decline | Stable |
| Hermit Thrush (*C. guttatus*) | HETH | – | – | 2009 | 1553 | Stable | Stable | Decline | Stable | Stable |
| Wood Thrush (*Hylocichla mustelina*) | WOTH | – | 10 | 207 | 385 | Decline | Decline | Decline | Decline | – |
| American Robin (*Turdus migratorius*) | AMRO | – | – | 7262 | 1179 | Decline | Decline | Decline | Decline | Stable |
| Gray Catbird (*Dumetella carolinensis*) | GRCA | – | 1 | 22923 | 17533 | Stable | Decline | Increase | Decline | – |
| Northern Mockingbird (*Mimus polyglottos*) | NOMO | – | – | 512 | – | Decline | NA | Decline | – | – |
| Brown Thrasher (*Toxostoma rufum*) | BRTH | 1 | – | 313 | 429 | Decline | Decline | Decline | Decline | – |
| Cedar Waxwing (*Bombycilla cedrorum*) | CEDW | – | – | 507 | 471 | Decline | Decline | Increase | Stable | Stable |
| Blue-winged Warbler (*Vermivora cyanoptera*) | BWWA | – | – | 218 | – | Decline | NA | Decline | – | – |
| Tennessee Warbler (*Oreothlypis peregrina*) | TEWA | – | – | 214 | 125 | Decline | Decline | – | Decline | Stable |
| Nashville Warbler (*Oreothlypis ruficapillus*) | NAWA | – | – | 665 | – | Stable | NA | Decline | Decline | Stable |
| Northern Parula (*Setophaga americana*) | NOPA | – | 4 | – | 366 | NA | Stable | Increase | Increase | Increase |
| Yellow Warbler (*S. petechia*) | YEWA | 2 | – | 250 | 874 | Decline | Stable | Stable | Decline | Decline |
| Magnolia Warbler (*S. magnolia*) | MAWA | 5 | – | 814 | 3380 | Decline | Increase | – | Stable | Increase |
| Cape May Warbler (*S. tigrine*) | CMWA | – | – | 469 | – | Decline | NA | – | Decline | Decline |
| Black-throated Blue Warbler (*S. caerulescens*) | BTBW | – | – | 684 | 567 | Stable | Stable | Stable | Increase | Increase |
| Yellow-rumped (Myrtle) Warbler (*S. coronata coronate*) | MYWA | 1 | – | 21014 | 754 | Decline | Stable | Stable | Stable | Stable |
| Black-throated Green Warbler (*S. virens*) | BTNW | – | 4 | 353 | 219 | Stable | Stable | Stable | Stable | Stable |
| Prairie Warbler (*S. discolor*) | PRAW | – | – | – | 142 | NA | Decline | Decline | – | – |
| Yellow Palm Warbler (*S. palmarum hypochrysea*) | YPWA | – | 4 | – | 342 | NA | Increase | – | Increase | Stable |
| Western Palm Warbler (*S. palmarum palmarum*) | WPWA | 2 | – | 286 | – | Decline | NA | – | – | – |
| Bay-breasted Warbler (*S. castanea*) | BBWA | – | – | 726 | – | Decline | NA | – | Decline | Stable |

*(Continued)*

**Table 1.** (*Continued*)

| Species | Code | Cluster+ | | Number Caught | | Overall Trends | | BBS Trends+ | | |
|---|---|---|---|---|---|---|---|---|---|---|
| | | Fall | Spr | Fall | Spr | Fall | Spr | NEMA | ANF | BSS |
| Blackpoll Warbler (*S. striata*) | BLPW | 1 | 3 | 7718 | 1113 | Decline | Decline | – | Decline | – |
| Black-and-white Warbler (*Mniotilta varia*) | BAWW | – | 1 | 1129 | 2442 | Decline | Decline | Decline | Decline | Stable |
| American Redstart (*S. ruticilla*) | AMRE | – | – | 3943 | 2965 | Decline | Decline | Stable | Decline | Stable |
| Ovenbird (*Seiurus aurocapilla*) | OVEN | 7 | – | 700 | 1512 | Stable | Stable | Decline | Increase | Stable |
| Northern Waterthrush (*Parkesia noveboracensis*) | NOWA | – | 3 | 922 | 1330 | Decline | Decline | Stable | Decline | Increase |
| Mourning Warbler (*Geothlypis philadelphia*) | MOWA | 5 | 2 | 366 | 535 | Stable | Decline | – | Decline | Stable |
| Common Yellowthroat (*G. trichas*) | COYE | 1 | 1 | 2125 | 4658 | Decline | Decline | Decline | Decline | Decline |
| Wilson's Warbler (*Cardellina pusilla*) | WIWA | 6 | – | 880 | 822 | Decline | Decline | – | Decline | Stable |
| Canada Warbler (*C. canadensis*) | CAWA | – | – | 496 | 1466 | Decline | Decline | Decline | Decline | Stable |
| Yellow-breasted Chat (*Icteria virens*) | YBCH | 6 | – | 1121 | – | Decline | NA | Decline | – | – |
| Eastern Towhee (*Pipilo erythrophthalmus*) | EATO | 2 | 6 | 893 | 1525 | Decline | Decline | Decline | Decline | – |
| Field Sparrow (*Spizella pusilla*) | FISP | 2 | – | 275 | – | Decline | NA | Decline | Decline | – |
| Song Sparrow (*Melospiza melodia*) | SOSP | 5 | 7 | 3107 | 877 | Decline | Stable | Decline | Decline | Decline |
| Lincoln's Sparrow (*M. lincolnii*) | LISP | 3 | – | 216 | 324 | Decline | Stable | – | Decline | Stable |
| Swamp Sparrow (*M. georgiana*) | SWSP | – | 7 | 1160 | 1341 | Stable | Stable | Decline | Increase | Stable |
| White-throated Sparrow (*Zonotrichia albicollis*) | WTSP | – | 1 | 8563 | 7038 | Decline | Decline | Decline | Decline | Decline |
| Dark-eyed (Slate-colored) Junco (*Junco hyemalis hyemalis*) | SCJU | 1 | 8 | 2237 | 241 | Decline | Stable | Stable | Decline | Stable |
| Northern Cardinal (*Cardinalis cardinalis*) | NOCA | – | 5 | 1892 | 747 | Increase | Increase | Increase | Increase | – |
| Red-winged Blackbird (*Agelaius phoeniceus*) | RWBL | – | – | – | 818 | NA | Decline | Decline | Decline | Decline |
| Common Grackle (*Quiscalus quiscula*) | COGR | – | – | – | 1543 | NA | Stable | Decline | Decline | Stable |
| Brown-headed Cowbird (*Molothrus ater*) | BHCO | – | 8 | – | 393 | NA | Decline | Stable | Decline | Decline |
| Baltimore Oriole (*Icterus galbula*) | BAOR | 2 | 6 | 774 | 1033 | Stable | Decline | Decline | Decline | – |
| American Goldfinch (*Spinus tristis*) | AMGO | – | – | 455 | 848 | Stable | Decline | Increase | Stable | Stable |
| Scarlet Tanager (*Piranga olivacea*) | SCTA | – | – | 231 | – | Stable | NA | Decline | Decline | Stable |
| Purple Finch (*Haemorhous purpureus*) | PUFI | 2 | – | 718 | – | Decline | NA | Decline | Decline | Stable |

+Cluster indicates which abundance trend cluster the species was sorted into by hierarchical clustering, if any. BBS Trends include the trend from three Breeding Bird Survey Regions: New England / Mid-Atlantic (NEMA), Atlantic Northern Forests (ANF), and Boreal Softwood Shield (BSS) for all birds with medium or high survey confidence in the region in question.

calculating effort-hours for a given species. For example, 98% of all spring Ovenbird captures across all years occurred between May 1 and June 5; therefore, all sampling efforts from outside this period were excluded when calculating the total spring effort-hours for Ovenbirds.

Once we had calculated the effort windows for each species, we converted the number of individuals caught in each year to the number of individuals caught / 10,000 effort-hours, in order to control for differing numbers of net-hours across years.

## State-space modeling

Because migration counts from a single site only sample a small portion of the population, and such counts are susceptible to the effects of weather [35], we used a state-space modeling approach to estimate the underlying trends in the actual size of each species' migratory cohort at Manomet. A state-space modeling approach allows us to separate the process variation (differing numbers of birds migrating through each year) from the observation variation (different proportions of those birds being caught each year) [36]. Our model was adapted from the one used by Kéry and Schaub [37], with the effort-adjusted number of birds caught in the first year

of reliable survey data (1969 for fall, 1970 for spring) as the prior for initial size of the migratory cohort. All models were run 200,000 times, with the first 100,000 runs discarded as a burn-in, using WinBUGS through R and the R2WinBUGS package [38, 39].

Once the models were complete, we made a coarse assessment of each species' net change in migratory cohort size across the study period by comparing the bird's estimated abundance in 2015 with the 95% confidence interval around its abundance in the first year of data (1969 for fall, 1970 for spring). Birds whose 2015 abundance exceeded the first year's upper 95% CI were classified as having significantly increased since the first year, while birds whose 2015 abundance was less than the first year's lower 95% CI were classified as having significantly declined. We then compared these qualitative trends to the trend estimates provided by the USGS Breeding Bird Survey for the New England Mid-Atlantic Region (BCR 30), the Atlantic Northern Forests Region (BCR 14), and the Boreal Softwood Shield Region (BCR 8) [40], since these regions are the likeliest breeding grounds of birds caught at Manomet. We only used BBS trend estimates for species that had Medium or High Regional Credibility in a given region.

## Cluster analysis of population patterns

With state-space model patterns in migratory cohort size over time already calculated, we were interested to determine if species could be grouped by their patterns of abundance over time. Accordingly, we standardized each species' time series to its own maximum value, thus preserving the shape of the overall trend and allowing us to compare species on the basis of trend shape alone. We clustered our species within each season (fall and spring) using the hierarchical Ward's method [41] as implemented in the R package pvclust [42]. Each point in a species' standardized time series was compared against equivalent points in each of the other time series, and the algorithm minimizes the Euclidean distances between time series to form clusters of similar trends. The pvclust packages identifies clusters that are statistically supported at the $p < 0.05$ level and creates a dendrogram.

In order to determine whether these clusters aligned with life-history traits among species, we classified all species according to several life-history traits that are frequently cited as being of conservation relevance [34,43]: dependence on open habitats (e.g., grassland and shrubland), nesting on or very near the ground, human commensals (frequently visit bird feeders and/or especially thrive in human-altered habitats), and being a spruce budworm (*Choristoneura* sp.) specialist. We also included different migratory strategies, since several studies have suggested that birds with longer migrations may be adjusting their migratory behavior less, and may fare especially poorly in response to climate change [44, 45]. Migratory strategies included: Resident (non-migratory), Facultative migrant (individuals within the same population may or may not migrate), and likely predominant wintering location: Southeastern United States, Caribbean, Central America, and South America. Many species belonged to more than one category of migratory strategy, but species were assigned to categories judged to be most representative of the migratory behavior of birds caught at Manomet. For a complete list of species life-history traits by species, see the Supplemental Material (S1 Table). We also calculated the median date of passage for each species in each season, with the assumption that migration timing might be a surrogate for a suite of possible ecological factors not covered by the other traits (e.g., distance migrated, which might be associated with the potential for phenological disruption [46–48]).

We then used these life-history traits in a k-modes clustering approach to sort all species into an equal number of clusters as in the time-series analysis (seven clusters for fall, ten clusters for spring). If membership in a particular time-series cluster is driven primarily by life-history traits, then we would expect the life-history clusters and time-series clusters to show high

concordance. For example, we might expect to see high concordance between a cluster of steadily declining time-series trends and a life-history cluster of open habitat specialists who migrate to South America. Concordance was assessed using multinomial logistic regression, with membership in a life-history cluster as a predictor variable and membership in a time-series cluster as a response variable. These results were compared to a null model (time-series cluster membership is random) and a model using median date of passage as a predictor variable.

## Results

In total, we analyzed information from 253,265 birds caught across 1,487,999 net hours during fall migration, and 172,609 birds captured across 925,327.5 net hours during spring migration (Table 1). The average 98% migration window was longer in the fall (65.6 days, ± 16.8) than in the spring (40.5 days, ± 13.2) (Table 1), as might be expected based on selective pressures for early arrival for breeding [49,50]. For the 43 species in our samples that appeared in both the fall and spring counts, the differences in migration windows are likely driven by the larger sample size and longer banding season in the fall.

Each of the season-specific state-space model graphs for every species is found in the Supplemental Materials. State-space model trends removed much of the interannual variation of raw time-series data (see example in Fig 1), but trends over time were often nonlinear and occasionally complex, defying easy categorization. Of the 62 fall species evaluated, 30 sorted into seven statistically significant clusters based on their time-series trends (Fig 2, Table 1). Group sizes ranged from 2 to 9 species. Cluster significance was determined by an approximately unbiased (AU) p-value < 0.05 from multiscale bootstrap resampling [42]. The clusters show that some species declined sharply in the late 1970s and then persisted at lower abundance (such as Eastern Towhee, Field Sparrow, Purple Finch, and Baltimore Oriole in cluster 2) while others have declined steadily over time (e.g., Magnolia Warbler, Mourning Warbler, Red-eyed Vireo and Song Sparrow in cluster 6). Others have increased overall, such as Blue-headed Vireo and Eastern Phoebe in cluster 4. (Fig 3; S1 Table). Some species showed a fair amount of interannual variation, but overall had no distinct net change over time (e.g., Blue-gray Gnatcatcher and Ovenbird in cluster 7).

Of the 52 spring species evaluated, 26 sorted into ten statistically significant clusters based on their time-series trends, with 2–5 species within each cluster (Fig 4). As with the fall clusters, different clusters of declining species exhibit distinct timing in the onset of decline. Birds in cluster 2 (Mourning Warbler and Least Flycatcher) showed a steep crash in spring captures in the late 1990s, while spring cluster 7 (Song Sparrow and Swamp Sparrow) had their greatest declines before 1980, and have since been stable or even recovering (Fig 5). Spring cluster 4 (Black-throated Green Warbler, Northern Parula, and Yellow Palm Warbler) are primarily united by an incredibly high rate of spring captures in 2010. Concordance in cluster membership between spring and fall clusters was remarkably low (Table 1).

For fall trends, 41 species (66%) showed significant declines in migratory cohort size between 1969 and 2015 (Table 1). Six species (10%) showed significant increases in migratory cohort size, and 15 species did not have a significantly different abundance in 2015 than in 1969. For spring migrants, 32 species (62%) showed a decline in abundance at Manomet since 1970, while six species (12%) increased significantly, and 14 species (27%) neither increased nor declined overall from 1970 to 2015. Several species demonstrated notable peaks and dips in abundance during the intervening years (see fall cluster 7 in Fig 4 for an example of this).

The group (cluster) affiliations of species based on life-history characteristics showed no concordance with cluster affiliations based on patterns of population size over time (Table 2).

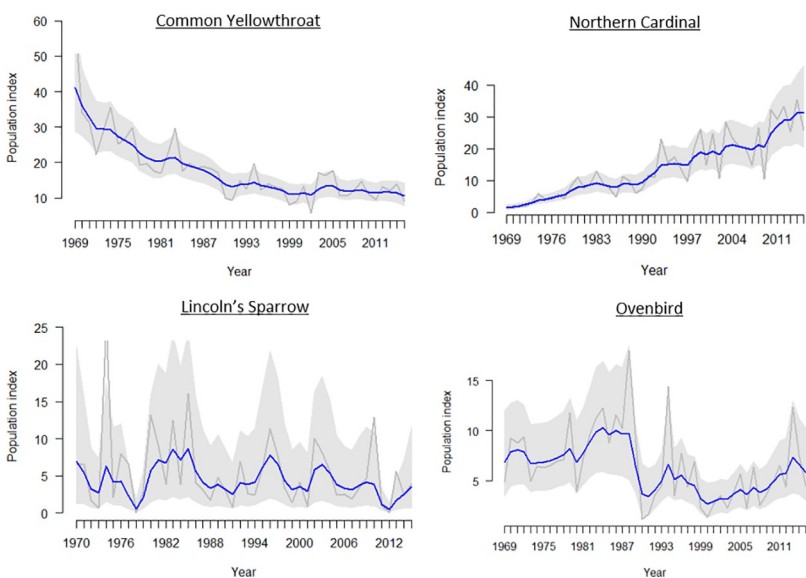

**Fig 1. Example time-series graphs of bird captures at Manomet for four different species from four different trend clusters.** Grey lines indicate raw capture data, blue lines indicate state-space estimates of actual migratory cohort size, and the shaded area is the 95% confidence interval around the state-space model estimate. Some species are unambiguously increasing or declining, while others show more complicated patterns. Lincoln's Sparrow data is from spring migration, all others are from fall.

This was true for both fall and spring, where their models never had support over the null model. The same is true for median passage date for fall birds, but not for spring birds. In spring, median date of passage was the best-supported model (Table 2), explaining a moderate amount of the variation in species membership between the two cluster types (maximum likelihood pseudo-$r^2$ = 0.56).

Our clustering analyses that evaluated species similarities in population size change over time left 47 fall species and 47 spring species not affiliated with any cluster. Of these, 27 species were found unclustered in both seasons.

## Discussion

In general, the patterns of abundance observed at Manomet signal that many of our landbird species are in trouble. With more than 60% of all species in both fall and spring showing significant declining trends, and fewer than 15% apparently increasing, our data support the widespread conservation concern that has hovered around North American migratory songbirds for decades [7,51]. That said, the lack of association between our trend clusters and life history traits suggests that simple narratives about the species most vulnerable to decline might not suffice.

For example, aerial insectivores are frequently cited as an avian group particularly at risk for decline [52,53]. Indeed, our data and the Breeding Bird Survey agree that Least Flycatchers and Eastern Wood-Pewees are probably declining in northeastern North America, yet the signals for Yellow-bellied Flycatchers are decidedly mixed, and Eastern Phoebes and Great Crested Flycatchers appear to be stable or increasing. Eastern Phoebes are short-distance migrants, but Great Crested and Yellow-bellied Flycatchers are Neotropical wintering birds just as Least Flycatcher is, and all use a wide variety of forested and second-growth habitats on their shared Central American wintering grounds [54–56]. Least Flycatchers are certainly declining in the eastern portion of their range, but the performance of other flycatchers

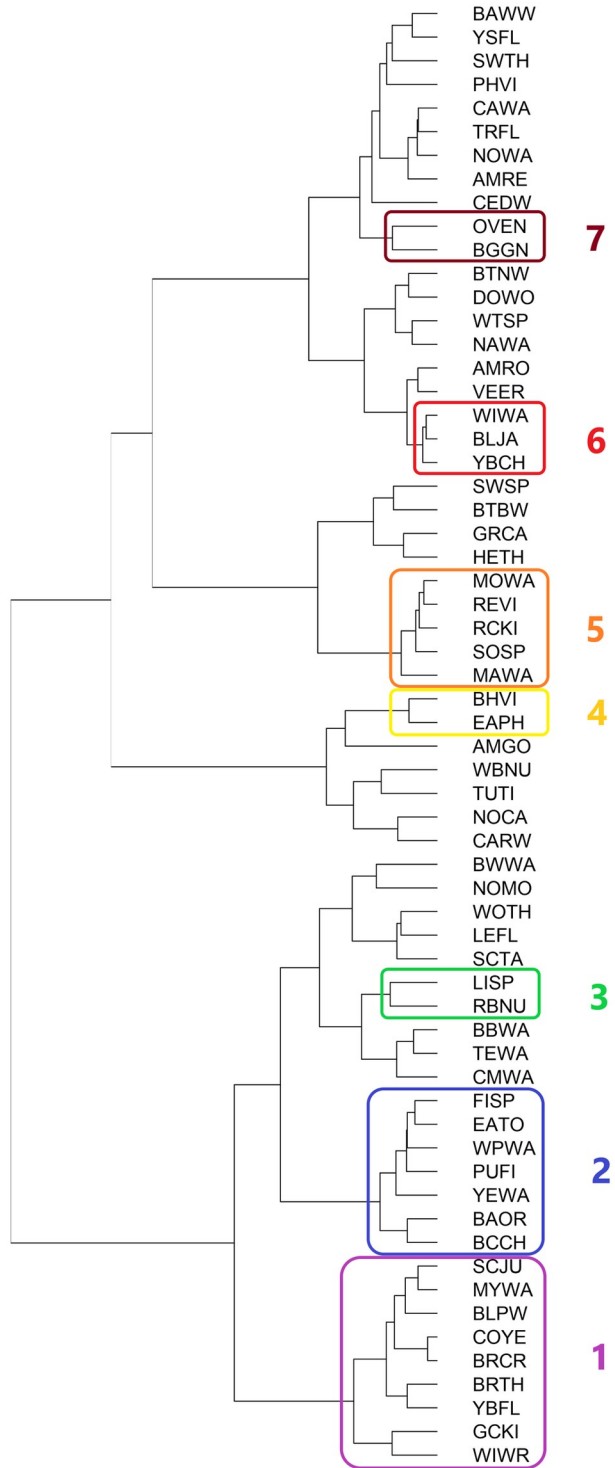

**Fig 2. Dendrogram of fall bird species, clustered via Ward's hierarchical clustering with a Euclidean distance method based on time series trend shape.** Colored rectangles enclose clusters significant at the approximately unbiased (AU) $p < 0.05$ level. 4-letter species codes and cluster numbers at the right of the figure match those in Table 1.

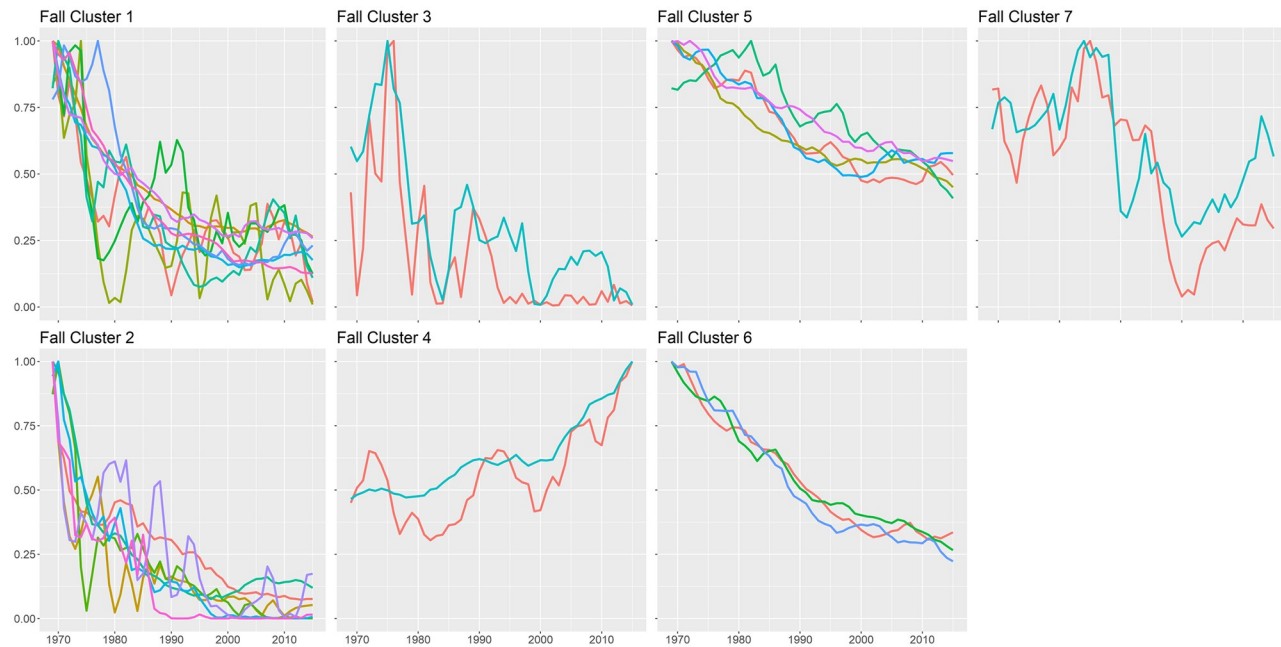

**Fig 3. Graphs of abundance trends over time of each significant fall species cluster.** Each colored line is a different species.

suggests that the explanation cannot be as simple as "loss of aerial insect food" or "loss of habitat in the Neotropics."

Magnolia Warbler is another interesting case. Declining at Manomet in the fall but increasing in the spring, the Breeding Bird Survey indicates that breeding populations north of Massachusetts are stable or increasing. It may be the case that Magnolia Warblers coming north in the spring represent a mixture of many breeding populations that separate somewhere north of Manomet. Fall birds may be primarily hatch-year birds from the breeding populations northeast of Manomet in New Brunswick and Nova Scotia, which are locally declining according to the Breeding Bird Survey. Magnolia Warblers are known to take more eastern routes in fall than in spring [57], and this serves as a good example of how migration capture data can supplement breeding-season surveys to complete an otherwise puzzling narrative. Such "loop migration" has also been demonstrated in Blackpoll Warblers. Via stable isotope analysis, northward spring migrant Blackpolls at Manomet have been linked to breeding populations east of Hudson Bay, while fall migrant Blackpolls that congregate at Manomet before crossing the Atlantic appear to be from western breeding populations [58].

Similarly, birds exhibiting similar trends over time are not necessarily responding to the same threats. Blue Jay and Wood Thrush, for example, show remarkably similar declines in spring abundance in cluster 10 (Fig 5). Aside from being broadly associated with "forests," however, these birds have almost nothing in common. They have different diets, different migration routes, different wintering grounds, and different nesting habits [59,60]. Yet, both are significantly declining in the New England region in both Manomet captures and Breeding Bird Survey abundance (Table 1). Vulnerability to cowbird parasitism, complex edge dynamics, and loss of Neotropical wintering habitat have all been implicated in Wood Thrush declines [59], but these seem unlikely to be major factors in the decline of Blue Jays [61]. Finally, discrepancies between patterns of abundance at Manomet and apparent trends in the Breeding Bird Survey underscore the complexity of avian population dynamics. Black-capped

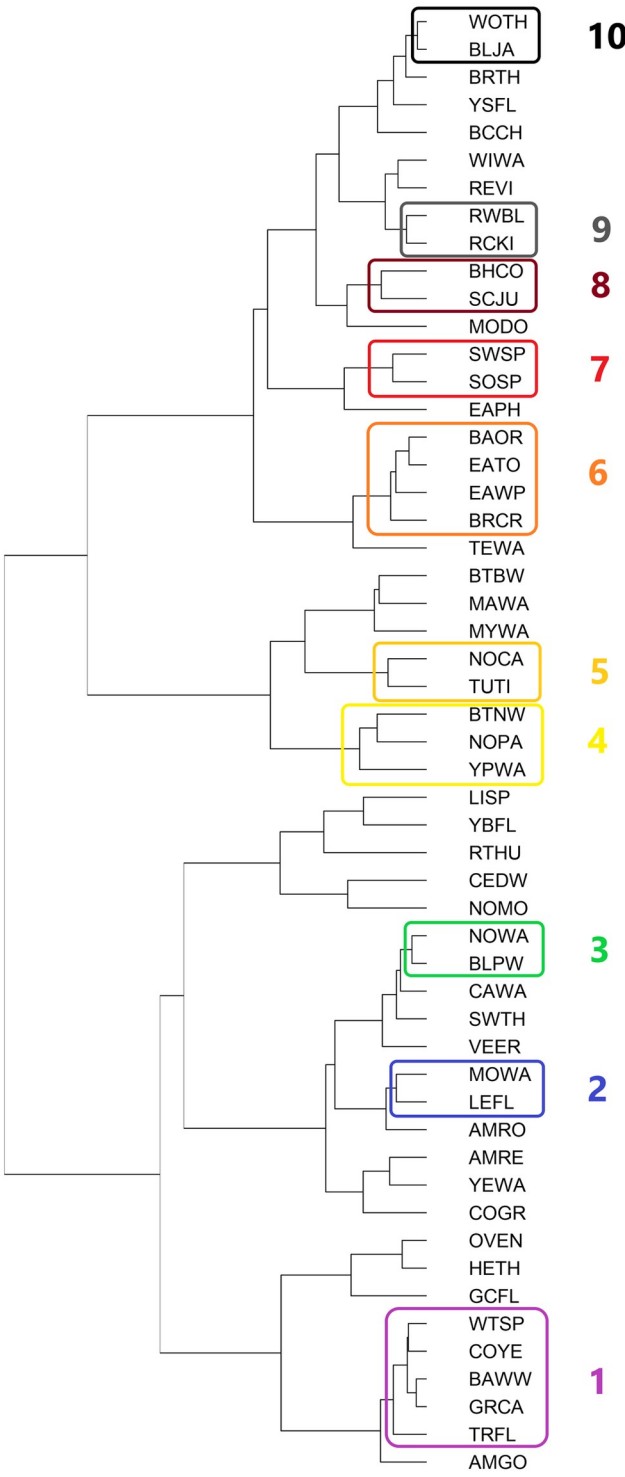

**Fig 4. Dendrogram of spring bird species, clustered via Ward's hierarchical clustering with a Euclidean distance method based on time series trend shape.** Colored rectangles enclose clusters significant at the approximately unbiased (AU) p < 0.05 level. 4-letter species codes and cluster numbers on the right of the figure match those in Table 1.

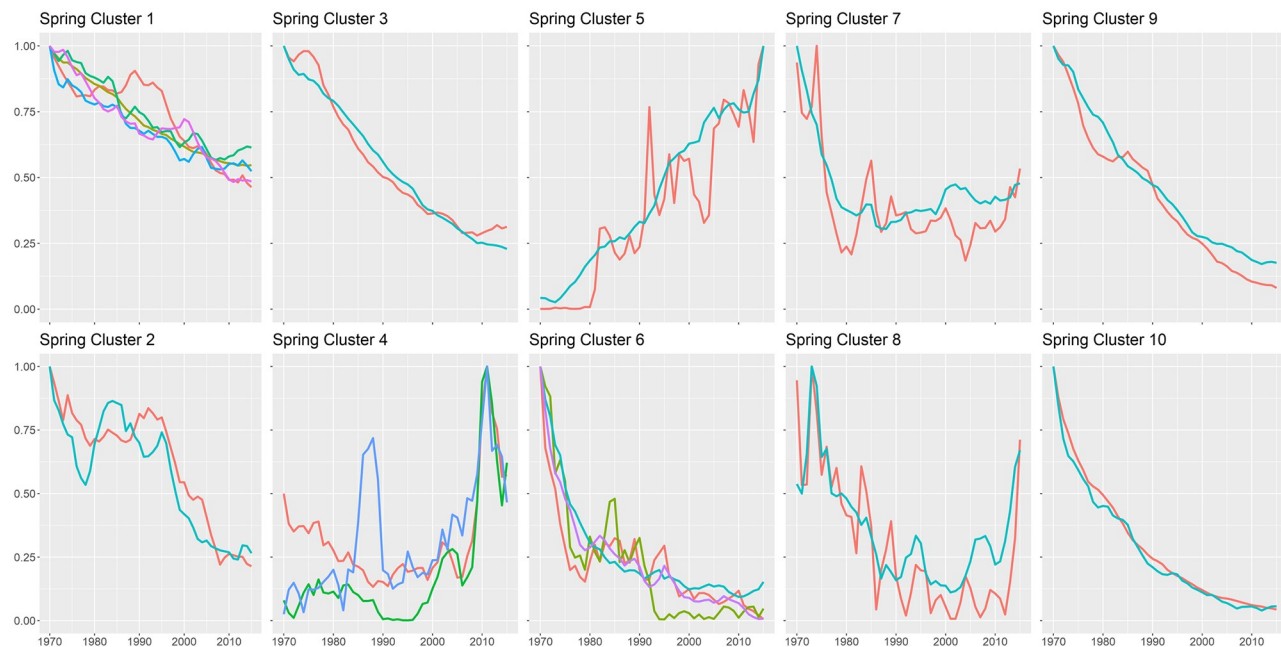

**Fig 5. Graphs of abundance trends over time of each significant spring species cluster.** Each colored line is a different species.

Chickadees are non-migratory and one of the most frequently caught birds at Manomet. In both fall and spring, chickadee captures have declined considerably since banding started at Manomet. The Breeding Bird Survey, by contrast, shows chickadee populations are stable or increasing in all regions nearby. Closer examination of the Black-capped Chickadee fall trends (S2 Table) shows that the apparent decline is driven by several "spikes" in fall captures during 1971 and the 1980s. Breeding seasons with high fledging success followed by a poor autumn seed crop have been shown to lead to these irruptive movements in an otherwise non-migratory species [62]. An increase in winter bird feeding by humans and milder winters as a result of climate change may have resulted in fewer large-scale autumn movements of Black-capped Chickadees, but determining the true population trend of northeastern chickadees from these data is not a straightforward enterprise. Apparent fall declines at Manomet in other species with short or facultative migration, such as Yellow-rumped Warbler, may similarly reflect changes in migratory behavior rather than actual population declines [63], but more complex dynamics may also be at play.

**Table 2. Multinomial modeling results.**

| Model (fall) | k | ΔAICc | ω | Model (spring) | k | ΔAICc | ω |
|---|---|---|---|---|---|---|---|
| Null model | 6 | 0.0 | 0.946 | Median arrival date | 18 | 0.0 | 0.70 |
| Median arrival date | 12 | 5.9 | 0.048 | Null model | 9 | 3.3 | 0.13 |
| Life history | 12 | 10.3 | 0.006 | Life history + median arrival | 27 | 3.4 | 0.13 |
| Life history + median arrival | 18 | 17.0 | <0.001 | Life history | 18 | 5.6 | 0.04 |

Results of multinomial models relating species affiliations with clusters based on patterns of change with the same number of clusters based on life-history characteristics (dependence on open habitats, nesting on or very near the ground, human commensals, and whether or not the species was a spruce budworm specialist). Degrees of freedom (k), differences in Akaike Information Criterion corrected for small sample size (ΔAICc), and model weights (ω) are reported.

Even so, for many species, trends at Manomet and those reported by the Breeding Bird Survey are in agreement, and the overall picture is a troubling one. Our findings on the prevalence of landbird declines are consistent with those of earlier analyses of Manomet data [31] as well as data from other northeastern migration sites [64,65]. Many migrant songbirds are showing significant long-term declines in migratory cohort size. This phenomenon is evident in Neotropical migrants, but also in many common and familiar species that migrate only short distances, such as American Robin and Blue Jay. Many of the species that show significant increasing trends in Manomet capture rates are resident human commensal species, such as Tufted Titmouse, Northern Cardinal, and White-breasted Nuthatch. Other increasing birds like Yellow Palm Warbler and Blue-headed Vireo are likely individuals wintering in the Gulf states of the USA rather than in the Neotropics.

Generally speaking, of the 28 species that did not fall into any significant clusters in either season, many had nearly horizontal abundance trendlines over time, either broadly stable (e.g., Hermit Thrush, S1 Table) or slowly declining (e.g., American Robin, Canada Warbler, S1 Table). Many were species that Manomet catches in relatively small numbers (e.g., Ruby-throated Hummingbird, Yellow-bellied Sapsucker, Great Crested Flycatcher). There were no obvious life history traits in common among these species, although smaller groups of birds with commonalities do exist within the group. A few (Bay-breasted Warbler, Cape May Warbler, and Tennessee Warbler) showed trends clearly driven by spruce budworm outbreaks in the 1970s [66], and a few exhibited strange patterns that defy easy description but may be related to unpredictable captures of wandering foraging flocks in late fall (e.g., American Goldfinch, Cedar Waxwing; see S1 Table).

Interestingly, the large proportion of bird trend clusters defy simple mechanistic categorization. Some life-history traits were consistent with time-series groupings; for example, the spring significant trend cluster of Northern Cardinal and Tufted Titmouse accurately reflects the similar life histories of these resident seed-eating backyard birds. However, there are no apparent connections between species in many of the other clusters. The poor ability of life-history traits to predict time-series trend cluster membership strongly suggests that there are many "paths" to the same apparent abundance trajectory, as shown by the example of Blue Jay and Wood Thrush, above.

Even the largest clusters of species show few (or no) commonalities in life history. Fall cluster 1 (Yellow-bellied Flycatcher, Brown Creeper, Winter Wren, Golden-crowned Kinglet, Brown Thrasher, "Myrtle" Warbler, Blackpoll Warbler, Common Yellowthroat, and Dark-eyed Junco) are united by the fact that they all declined sharply in abundance at Manomet through the mid-1980s, and then leveled off to slower declines or near-stability. The closest apparent unifying life-history trait among them is association with mature conifer-dominated forests for breeding. However, two of these species (Brown Thrasher and Common Yellow-throat) do not use mature conifer woods at all, instead preferring disturbed areas of brushy shrubland for breeding, and other species that are strongly associated with breeding in conifer woodlands (such as Blue-headed Vireo and Black-throated Green Warbler) did not associate with this cluster.

The predictive utility of median date of migration in predicting spring trends underscores a point made by previous researchers about the importance of weather in the movements of migrating birds [30,67]. Large movements can occur with little warning as conditions change, and birds with similar migratory timing may be caught in large numbers or missed altogether depending on whether nets are open during a major migratory fallout [68]. Spring migration, especially, is a time when migrating birds are attempting to return to the breeding grounds as quickly as possible in order to secure high-quality territories and mates [69]. For example, as previously mentioned, spring trends for Blue Jay and Wood Thrush align very neatly and

clustered significantly in the hierarchical analysis (Fig 5), but in terms of life history, these two birds share very little ecologically beyond being "forest birds" in the broadest possible sense. However, their peak spring migration dates are the same, and so their annual spring capture rates were likely strongly influenced by weather during peak migration. That both birds are declining is supported by their Manomet abundance trends and by the Breeding Bird Survey [13], but again, it is likely that they are declining for different reasons.

As with any migration study based on data from a single site, even if that site is drawing birds from larger breeding and wintering ranges, these results should be interpreted with care. Birds captured at Manomet do not compose a random sample of any species' populations, but in many cases they integrate data from several breeding populations, as well as from birds whose breeding grounds are too remote for effective surveys by other methods [30]. Recent work from Long Point Bird Observatory and Powdermill Avian Research Center has confirmed that banding stations tend to capture fall juvenile birds out of proportion with their abundance in the population [70]. Manomet's coastal location probably amplifies this effect, especially in the fall [23], since younger inexperienced birds are more likely to become disoriented or exhausted at coastal sites and land there [71]. Where fall and spring apparent trends differed, as in 13 of 73 (18%) of the species analyzed here, such differences are likely attributable to 1) differences in fecundity (fall) and overwinter survival (spring), 2) differences in the breeding populations being sampled in each season, as for Magnolia Warbler or Blackpoll Warbler, 3) species taking different migratory routes in each season, and 4) differences in age structure of migratory populations sampled at coastal sites. In this way, as long as the data are interpreted with a careful understanding of their limitations, insights can be drawn from migratory data that breeding and wintering surveys alone cannot provide.

In an era of rapid global change, studies using migration data can detect potential behavioral shifts such as those of Black-capped Chickadees and Yellow-rumped Warblers, above. Such population-wide changes in migratory behavior in wild birds have already been observed [72,73]. As winters in northern North America grow milder and storms become more unpredictable, it seems likely that short-distance and facultative migrant species will overwinter farther north than in the past, or in some cases, not migrate at all.

## Supporting information

**S1 Table. This table indicates which life history traits were assigned to each species, which cluster that species was sorted into in fall and spring based on life history traits (lifehist. num.f and lifehist.num.s), which statistically significant cluster that species was sorted into in fall and spring (trend.f and trend.s), the start date, end date, and length (in days) of each species' window of passage in fall and spring, median date of passage (in ordinal day of year) in fall and spring, and an estimate of the species' apparent trend in fall and spring abundance at Manomet over the full study period.**
(XLSX)

**S2 Table. The number of new fall captures of each species by Institute for Bird Populations (IBP) four-letter code for each year from 1969–2015.** NET.HOURS indicates the total number of effort-hours for that season.
(CSV)

**S3 Table. The number of new spring captures of each species by Institute for Bird Populations (IBP) four-letter code for each year from 1970–2015.** NET.HOURS indicates the total number of effort-hours for that season.
(CSV)

## Acknowledgments

We thank the tremendous number of student and volunteer banders who contributed thousands of hours banding birds at Manomet from the 1960s through today. The trustees and friends of Manomet provided constant logistical and financial support to this important work.

## Author Contributions

**Conceptualization:** Matthew D. Kamm, Trevor L. Lloyd-Evans, Maina Handmaker, J. Michael Reed.

**Data curation:** Trevor L. Lloyd-Evans.

**Formal analysis:** Matthew D. Kamm, J. Michael Reed.

**Investigation:** Matthew D. Kamm, Trevor L. Lloyd-Evans, Maina Handmaker.

**Methodology:** Matthew D. Kamm, Trevor L. Lloyd-Evans.

**Project administration:** Matthew D. Kamm.

**Resources:** Trevor L. Lloyd-Evans.

**Supervision:** J. Michael Reed.

**Writing – original draft:** Matthew D. Kamm, Maina Handmaker.

**Writing – review & editing:** Matthew D. Kamm, Trevor L. Lloyd-Evans, Maina Handmaker, J. Michael Reed.

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
