## [Decision Letter · Decision Letter 0]

8 Jul 2019

PONE-D-19-15287

A Half-century of Changes in Migratory Landbird Numbers along Coastal Massachusetts

PLOS ONE

Dear Dr. Kamm,

Thank you for submitting your manuscript to PLOS ONE. After careful consideration, we feel that it has merit but does not fully meet PLOS ONE’s publication criteria as it currently stands. Therefore, we invite you to submit a revised version of the manuscript that addresses the points raised during the review process.

Specific comments from the reviewers and me are given below.

We would appreciate receiving your revised manuscript by 20 August, 2019. To enhance the reproducibility of your results, we recommend that if applicable you deposit your laboratory protocols in protocols.io, where a protocol can be assigned its own identifier (DOI) such that it can be cited independently in the future. For instructions see: http://journals.plos.org/plosone/s/submission-guidelines#loc-laboratory-protocols

We look forward to receiving your revised manuscript.

Kind regards,

Brian G. Palestis, Ph.D.

Academic Editor

PLOS ONE

Journal Requirements:

1. Thank you for including the following funding information; "This work was internally funded."

Please provide an amended Funding Statement that declares *all* the funding or sources of support received during this specific study (whether external or internal to your organization) as detailed online in our guide for authors at http://journals.plos.org/plosone/s/submit-now.  

Please state what role the funders took in the study.  If any authors received a salary from any of your funders, please state which authors and which funder. If the funders had no role, please state: "The funders had no role in study design, data collection and analysis, decision to publish, or preparation of the manuscript."

Additional Editor Comments (if provided):

Both reviewers recommend minor revision and enjoyed your study. I agree with them. However, please go through their comments and make the suggested changes, or explain why not for those that you do not change.

I have pasted the review letter from Dick Veit, which came directly to me rather than the PLOS ONE site. The other review letter is uploaded below.

In addition to the comments from the reviewers, I also have a few minor comments:

1) You may want to check that the analyses also work with 1970 spring data excluded, as for 1969.

2) Check the k values in Table 2. Why are they so different for similar variables between fall and spring?

3) Please stick to one format for the list of references. You have several varieties, some of which are not normally used in scientific writing (e.g. giving issue numbers, capitalizing all words in an article title, etc.).

Veit's review:

Detailed comments below. I found this to be a very interesting paper, well suited to publication in PLoS One. I think that the use of long-term migration data for estimating population trends has been an underutilized resource, but clearly has its strengths as pointed out by the authors. Migration data are not without limitations (also pointed out by the authors) but especially in combination with BBS and CBC data provide a powerful tool. I like the use of the clustering techniques to try to identify what species cluster together on the basis of population trend, and the results are interesting to a broad audience interested on climate change and its impacts on any organisms –these results suggest a diversity of species-specific responses ; this broad appeal makes the paper particularly suitable to PLoS ONE (as opposed to an ornithology journal). I am not an expert on state space models but know enough about the approach and feel that the techniques used were appropriate, and interesting in that I have never seen them used in quite this context.

Furthermore, the MBO database on migration is truly extraordinary for its persistence and continuity since 1966, and that in itself lends credibility to the results of this study.

1.) With most species showing significant net declines in at least one season (49/73 species evaluated). How can a species decline in one season?

2.) Because banding stations are typically situated at narrow points in migration Please use different wording here or explain more fully. I think what you mean is that, since, during migration, birds of a variety of species breeding in a variety of localities tend to co-occur at the same places during migration, counts of migrants integrate population trends over broader spatial scales than is possible with censuses of breeding birds. If that is indeed what you mean, then I would emphasize this point more strongly here and in the abstract. Ornithologists have a bias against counts of migrants because they think such counts are too “messy”, whereas in some sense the opposite is true.

3.) For a more thorough account of local conditions, see (delete this phrase, top of page 6)

4.) Approval of the work by an ethics committee is not required for catching and banding

birds in the United States. Such requirement is legally determined by the institution concerned; I would delete if I were you!

5.) Because Willow Flycatcher and Alder Flycatcher cannot be separated reliably in the hand. I disagree, and think the evidence is strong that the majority can be identified (Pyle, latest edition). Do you have guesses? If so it seems there are changes going on with both species, and it would be interesting to identify migrants (and dates of migration). As a contrast to your next statement, I suspect that Willow and Alder flycatchers are more reliably separated than are subspecies of Palm Warblers.

6.) (distinguished by physiological signs of breeding

readiness, or hatch-year birds caught on spring migration). What do you mean by a HY bird here - I guess a < 1 yr old bird in may would be SY rather than HY but in a sense it is a “hatching year bird”. So you are talking about baby chickadees and the like that you capture during May? My first reaction was that you excluded all SY birds in spring, which I am sure is not true. Maybe phrase a bit more clearly.

7.) Page 7, second to last paragraph. How does excluding captures from outside the peak migration periods affect your results for species whose peak migration periods have changed (some quite substantially, I think)?

8.) being a spruce budworm (Choristoneura sp.) specialist. I wouldn’t argue with any species previously determined to be a spruce budworm specialist, but what is the basis for saying any given species is not? Might not any passerine nesting in boreal forests benefit from outbreaks of these caterpillars?

9.) What is your basis for saying that any species lacks plasticity in migration strategy? MØller is a proven fraud; I haven’t read Both 2010. I am suspicious, and think that most species are fairly plastic, given the “incentive” to be.

10.) Page 20 typo Blackpool Warblers

Dick Veit

Reviewers' comments:

Reviewer's Responses to Questions

**Comments to the Author**

1. Is the manuscript technically sound, and do the data support the conclusions?

Reviewer #1: Yes

2. Has the statistical analysis been performed appropriately and rigorously? 

Reviewer #1: Yes

3. Have the authors made all data underlying the findings in their manuscript fully available?

Reviewer #1: Yes

4. Is the manuscript presented in an intelligible fashion and written in standard English?

Reviewer #1: Yes

5. Review Comments to the Author

Reviewer #1: This is a well thought out paper that analyzes long-term banding data in an effort to determine trends in bird populations. The results and implications of the work are of great value, especially since there are many challenges in the interpretation of data from banding efforts during migration.

Complicating factors related to synthesizing data collected in this manner (e.g. mixing of populations of species from different regions with different population trends, weather patterns related to overall abundance observed as captures at a single location) are discussed sufficiently and effectively. And the fact that results of this study generally reflect population declines documented using different methodologies and study designs, lends support to the authors' conclusions and highlights the importance of publishing this paper in the peer reviewed literature.

A few comments below on how revising and expanding the interpretation of the data could provide additional insight and thoughts for follow up work:

1. A more expanded interpretation of each cluster would be useful. While some discussion of species in each cluster is provided, more detail on the species clustering can provide a better understanding for the reader about what the authors' thoughts are about species composition in each cluster.

2. In addition, are there any characteristics of the species that don't belong in any cluster?

3. There is no concordance in species clustering for spring and fall, and the authors explain that this as the result of potentially the result of sampling different breeding populations in different seasons or species using different migratory routes. While this may not be the broader purpose of the paper, looking at patterns more closely and providing a better in-depth understanding of the discrepancies would be useful.

4. Clusters consist of different number of species (2-9 species). It would interesting to understand if similarities in life history traits may be more strongly reflected in species belonging to larger clusters. In other words, while the lack of similarity in spring cluster 10 (Blue Jay and Wood Thrush) is discussed, are there stronger similarities in the larger groups(for example fall group 1). Do these greatly dissimilar clusters of 2 drive the lack of relation between species affiliation and cluster? That should be further explored.

Thanks for the opportunity to review a very nice project!

6. PLOS authors have the option to publish the peer review history of their article (what does this mean?). If published, this will include your full peer review and any attached files.

Reviewer #1: No

---

## [Author Response · Author response to Decision Letter 0]

16 Aug 2019

Response: We have revised our formatting to comply with PLOS ONE’s style requirements.

1. Thank you for including the following funding information; "This work was internally funded."

a. Please provide an amended Funding Statement that declares *all* the funding or sources of support received during this specific study (whether external or internal to your organization) as detailed online in our guide for authors at http://journals.plos.org/plosone/s/submit-now. 

b. Please state what role the funders took in the study. If any authors received a salary from any of your funders, please state which authors and which funder. If the funders had no role, please state: "The funders had no role in study design, data collection and analysis, decision to publish, or preparation of the manuscript."

Response: Our amended funding statement is as follows. This work was funded entirely by the generosity of Manomet’s friends, donors, and trustees. This funding paid the salaries of MK, MH, and TLE during their work on this project. The donors had no role in study design, data collection and analysis, decision to publish, or preparation of the manuscript. All data collection was done by Manomet employees or volunteers; design of this study, data analysis, decision to publish, and preparation of the manuscript was done by the authors with no input from any donor or trustee.

Additional Editor Comments (if provided):

Both reviewers recommend minor revision and enjoyed your study. I agree with them. However, please go through their comments and make the suggested changes, or explain why not for those that you do not change.

I have pasted the review letter from Dick Veit, which came directly to me rather than the PLOSONE site. The other review letter is uploaded below.

In addition to the comments from the reviewers, I also have a few minor comments:

1) You may want to check that the analyses also work with 1970 spring data excluded, as for 1969.

Response: We repeated the analyses with spring 1970 excluded, and found some changes in cluster membership. Some clusters remained the same, others gained a species, and some new significant clusters formed from species not previously included in any cluster. The same happened when we then excluded 1971, making spring 1972 the first year. Ultimately, the shape of the trend (which is the basis for the clusters) is undoubtedly sensitive to the starting values, and this will remain true regardless of which year is chosen as the first. We have re-checked the effort distribution and count data from 1970 and found the data quality to be consistent with that of the several following years. Therefore, in the interest of covering as many years as possible. we have elected to leave 1970 as the first year of spring coverage.

2) Check the k values in Table 2. Why are they so different for similar variables between fall and spring?

Response: We thank the editor for bringing this to our attention. This was due to an error in data structures, wherein median date of arrival in the spring was stored as a factor with 45 unique levels, rather than as a numeric quantity. This has been corrected, and the model ranks remained unchanged. Values in Table 2 have been updated accordingly.

3) Please stick to one format for the list of references. You have several varieties, some of which are not normally used in scientific writing (e.g. giving issue numbers, capitalizing all words in an article title, etc.).

Response: Revisions have been proofread and re-formatted to meet journal guidelines.

Veit's review:

Detailed comments below. I found this to be a very interesting paper, well suited to publication in PLoS One. I think that the use of long-term migration data for estimating population trends has been an underutilized resource, but clearly has its strengths as pointed out by the authors. Migration data are not without limitations (also pointed out by the authors) but especially in combination with BBS and CBC data provide a powerful tool. I like the use of the clustering techniques to try to identify what species cluster together on the basis of population trend, and the results are interesting to a broad audience interested on climate change and its impacts on any organisms –these results suggest a diversity of species-specific responses ; this broad appeal makes the paper particularly suitable to PLoS ONE (as opposed to an ornithology journal). I am not an expert on state space models but know enough about the approach and feel that the techniques used were appropriate, and interesting in that I have never seen them used in quite this context.

Furthermore, the MBO database on migration is truly extraordinary for its persistence and continuity since 1966, and that in itself lends credibility to the results of this study.

1.) With most species showing significant net declines in at least one season (49/73 species evaluated). How can a species decline in one season?

Revision made: With most species showing significant net declines in migratory cohort size in spring, fall or both (49/73 species evaluated). 

2.) Because banding stations are typically situated at narrow points in migration. Please use different wording here or explain more fully. I think what you mean is that, since, during migration, birds of a variety of species breeding in a variety of localities tend to co-occur at the same places during migration, counts of migrants integrate population trends over broader spatial scales than is possible with censuses of breeding birds. If that is indeed what you mean, then I would emphasize this point more strongly here and in the abstract. Ornithologists have a bias against counts of migrants because they think such counts are too “messy”, whereas in some sense the opposite is true.

Revision made: Because banding stations are typically situated in areas such as mountain gaps, desert oases, and coastal points where birds from many breeding populations naturally aggregate during migration…

Revision made: Migration counts are prone to large interannual variation and trends are often difficult to interpret, but have the advantage of sampling many breeding populations in a single locale

3.) For a more thorough account of local conditions, see (delete this phrase, top of page 6)

Revision made.

4.) Approval of the work by an ethics committee is not required for catching and banding

birds in the United States. Such requirement is legally determined by the institution concerned; I would delete if I were you!

Revision made.

5.) Because Willow Flycatcher and Alder Flycatcher were not separated reliably in the hand for the first three decades of this study. I disagree, and think the evidence is strong that the majority can be identified (Pyle, latest edition). Do you have guesses? If so it seems there are changes going on with both species, and it would be interesting to identify migrants (and dates of migration). As a contrast to your next statement, I suspect that Willow and Alder flycatchers are more reliably separated than are subspecies of Palm Warblers.

Revision made: Relevant wing formula data for the separation of Alder Flycatcher and Willow Flycatcher were not collected for more than half of the study period, thus we have adopted the parsimonious strategy of not separating these two species trends and both are presented as “Traill’s Flycatcher.”

Response: Although many Traill’s complex Empidonax can be identified with modern methods, we lack the recorded wing data to do so with more than half of the Traill’s records in this dataset. It has not been our experience that Willow and Alder flycatcher are more reliably separated than the Palm Warbler subspecies.

6.) (distinguished by physiological signs of breeding

readiness, or hatch-year birds caught on spring migration). What do you mean by a HY bird here - I guess a < 1 yr old bird in may would be SY rather than HY but in a sense it is a “hatching year bird”. So you are talking about baby chickadees and the like that you capture during May? My first reaction was that you excluded all SY birds in spring, which I am sure is not true. Maybe phrase a bit more clearly.

Revision made: distinguished by physiological signs of breeding readiness, or local fledgling birds caught during spring migration…

7.) Page 7, second to last paragraph. How does excluding captures from outside the peak migration periods affect your results for species whose peak migration periods have changed (some quite substantially, I think)?

Response: Since only 2% of all captures are excluded and the peak window is calculated across all years, only the very earliest and very latest records are excluded; the use of the full dataset to calculate peak migration makes the method sensitive to shifts in peak migration.

8.) being a spruce budworm (Choristoneura sp.) specialist. I wouldn’t argue with any species previously determined to be a spruce budworm specialist, but what is the basis for saying any given species is not? Might not any passerine nesting in boreal forests benefit from outbreaks of these caterpillars?

Response: The reviewer makes a good point; our definition is particular to birds that breed mostly or only in the boreal forest and whose abundance patterns at Manomet seem to be primarily determined by the abundance of spruce budworm in the preceding breeding season.

9.) What is your basis for saying that any species lacks plasticity in migration strategy? MØller is a proven fraud; I haven’t read Both 2010. I am suspicious, and think that most species are fairly plastic, given the “incentive” to be.

Revision made: …have suggested that birds with longer migrations may be adjusting their migratory behavior less, and may fare especially poorly in response to climate change (Butler 2003, Both et al. 2010)

Response: It is true that it is difficult to separate a lack of response from a lack of ability to respond; we have adjusted our language to reflect the widely-documented disparity in adjustment of migratory timing between short- and long-distance migrants. We have replaced the reference to Møller et al.

10.) Page 20 typo Blackpool Warblers

Revision made: Blackpoll Warblers

5. Review Comments to the Author

Reviewer #1: This is a well thought out paper that analyzes long-term banding data in an effort to determine trends in bird populations. The results and implications of the work are of great value, especially since there are many challenges in the interpretation of data from banding efforts during migration.

Complicating factors related to synthesizing data collected in this manner (e.g. mixing of populations of species from different regions with different population trends, weather patterns related to overall abundance observed as captures at a single location) are discussed sufficiently and effectively. And the fact that results of this study generally reflect population declines documented using different methodologies and study designs, lends support to the authors' conclusions and highlights the importance of publishing this paper in the peer reviewed literature.

A few comments below on how revising and expanding the interpretation of the data could provide additional insight and thoughts for follow up work:

1. A more expanded interpretation of each cluster would be useful. While some discussion of species in each cluster is provided, more detail on the species clustering can provide a better understanding for the reader about what the authors' thoughts are about species composition in each cluster.

Response: See response to point 4.

2. In addition, are there any characteristics of the species that don't belong in any cluster?

Revision made: Generally speaking, of the 28 species that did not fall into any significant clusters in either season, many had nearly horizontal abundance trendlines over time, either broadly stable (e.g., Hermit Thrush, Supplemental Material) or slowly declining (e.g., American Robin, Canada Warbler, Supplemental Material). Many were species that Manomet catches in relatively small numbers (e.g., Ruby-throated Hummingbird, Yellow-bellied Sapsucker, Great Crested Flycatcher). There were no obvious life history traits in common among these species, although smaller groups of birds with commonalities do exist within the group.

Response: We have expanded our discussion of this group of species.

3. There is no concordance in species clustering for spring and fall, and the authors explain that this as the result of potentially the result of sampling different breeding populations in different seasons or species using different migratory routes. While this may not be the broader purpose of the paper, looking at patterns more closely and providing a better in-depth understanding of the discrepancies would be useful.

4. Clusters consist of different number of species (2-9 species). It would interesting to understand if similarities in life history traits may be more strongly reflected in species belonging to larger clusters. In other words, while the lack of similarity in spring cluster 10 (Blue Jay and Wood Thrush) is discussed, are there stronger similarities in the larger groups(for example fall group 1). Do these greatly dissimilar clusters of 2 drive the lack of relation between species affiliation and cluster? That should be further explored.

Revision made: Even the largest clusters of species show few (or no) commonalities in life history. Fall cluster 1 (Yellow-bellied Flycatcher, Brown Creeper, Winter Wren, Golden-crowned Kinglet, Brown Thrasher, “Myrtle” Warbler, Blackpoll Warbler, Common Yellowthroat, and Dark-eyed Junco) are united by the fact that they all declined sharply in abundance at Manomet through the mid-1980s, and then leveled off to slower declines or near-stability. The closest apparent unifying life-history trait among them is association with mature conifer-dominated forests for breeding. However, two of these species (Brown Thrasher and Common Yellowthroat) do not use mature conifer woods at all, instead preferring disturbed areas of brushy shrubland for breeding, and other species that are strongly associated with breeding in conifer woodlands (such as Blue-headed Vireo and Black-throated Green Warbler) did not associate with this cluster.

Response: We appreciate these comments, and agree that there is much of interest in delving into the specific details of each group. However, it is our opinion that adding individual analysis for each of the seventeen significant clusters would add considerably to the length of the paper without adding significantly to the paper’s specific narrative; no groups had strong life history connections beyond those already mentioned in the text. We have added the above analysis of fall cluster 1 as an illustrative example; it is possible to propose explanations for cluster membership in each case, but every cluster includes exceptions to these explanations and lacks species that would be expected if the explanations were primary driving forces.

---

## [Editor Report · Decision Letter 1]

22 Aug 2019

PONE-D-19-15287R1

A half-century of changes in migratory landbird numbers along coastal Massachusetts

PLOS ONE

Dear Mr. Kamm,

Thank you for submitting your manuscript to PLOS ONE. After careful consideration, we feel that it has merit but does not fully meet PLOS ONE’s publication criteria as it currently stands. Therefore, we invite you to submit a revised version of the manuscript that addresses the points raised during the review process.

We would appreciate receiving your revised manuscript by Oct 06 2019 11:59PM. To enhance the reproducibility of your results, we recommend that if applicable you deposit your laboratory protocols in protocols.io, where a protocol can be assigned its own identifier (DOI) such that it can be cited independently in the future. For instructions see: http://journals.plos.org/plosone/s/submission-guidelines#loc-laboratory-protocols

We look forward to receiving your revised manuscript.

Kind regards,

Brian G. Palestis, Ph.D.

Academic Editor

PLOS ONE

Additional Editor Comments (if provided):

Thank you very much for the detailed responses to the questions from the reviewers and editors, and for the changes made to the manuscript.

Before accepting the article for publication, there are two questions I have that should be addressed:

1) I had asked "Check the k values in Table 2. Why are they so different for similar variables between fall and spring?" and your response was "We thank the editor for bringing this to our attention. This was due to an error in data structures, wherein median date of arrival in the spring was stored as a factor with 45 unique levels, rather than as a numeric quantity. This has been corrected, and the model ranks remained

unchanged. Values in Table 2 have been updated accordingly."

When I look at Table 2, I see only changes to the delta-AICc values and model weights but not to k. Is this correct? If not, then please correct the values. If yes, then please add a sentence to the caption explaining why.

2) The paper cited by Dorian et al. is not published (listed as "in review"). Is there a pre-print or something else you can cite? If you don't have more information, then this reference should be removed from your manuscript. If there is something more specific to cite, then the numbering for citations would also need to change.
---

## [Author Response · Author response to Decision Letter 1]

22 Aug 2019

Dear Handling Editors,

Thank you very much for the opportunity to revise our manuscript, “A half-century of changes in migratory landbird numbers along coastal Massachusetts. We have made both of the requested revisions. Our responses and revisions are detailed in the letter below, along with the feedback that prompted them.

1) I had asked "Check the k values in Table 2. Why are they so different for similar variables between fall and spring?" and your response was "We thank the editor for bringing this to our attention. This was due to an error in data structures, wherein median date of arrival in the spring was stored as a factor with 45 unique levels, rather than as a numeric quantity. This has been corrected, and the model ranks remained

unchanged. Values in Table 2 have been updated accordingly."

When I look at Table 2, I see only changes to the delta-AICc values and model weights but not to k. Is this correct? If not, then please correct the values. If yes, then please add a sentence to the caption explaining why.

Revision made: We have corrected the k values; this was missed in the last round of corrections.

2) The paper cited by Dorian et al. is not published (listed as "in review"). Is there a pre-print or something else you can cite? If you don't have more information, then this reference should be removed from your manuscript. If there is something more specific to cite, then the numbering for citations would also need to change.

Revision made: We have removed this reference.

---

## [Editor Report · Decision Letter 2]

26 Aug 2019

A half-century of changes in migratory landbird numbers along coastal Massachusetts

PONE-D-19-15287R2

Dear Dr. Kamm,

We are pleased to inform you that your manuscript has been judged scientifically suitable for publication and will be formally accepted for publication once it complies with all outstanding technical requirements.

With kind regards,

Brian G. Palestis, Ph.D.

Academic Editor

PLOS ONE

Additional Editor Comments (optional):

Thank you for making these changes.
---

## [Editor Report · Acceptance letter]

29 Aug 2019

PONE-D-19-15287R2 

A half-century of changes in migratory landbird numbers along coastal Massachusetts 

Dear Dr. Kamm:

I am pleased to inform you that your manuscript has been deemed suitable for publication in PLOS ONE. Congratulations! Your manuscript is now with our production department. 

With kind regards,

on behalf of

Dr. Brian G. Palestis 

Academic Editor

PLOS ONE